# Volatile scent chemicals in the urine of the red fox, *Vulpes vulpes*

**Stuart McLean**[1]*, **David S. Nichols**[2], **Noel W. Davies**[2]

**1** School of Pharmacy and Pharmacology, University of Tasmania, Hobart, Tasmania, Australia, **2** Central Science Laboratory, University of Tasmania, Hobart, Tasmania, Australia

* stuart.mclean@utas.edu.au

## Abstract

The red fox is a highly adaptable mammal that has established itself world-wide in many different environments. Contributing to its success is a social structure based on chemical signalling between individuals. Urine scent marking behaviour has long been known in foxes, but there has not been a recent study of the chemical composition of fox urine. We have used solid-phase microextraction and gas chromatography-mass spectrometry to analyze the urinary volatiles in 15 free-ranging wild foxes (2 female) living in farmlands and bush in Victoria, Australia. Foxes here are routinely culled as feral pests, and the urine was collected by bladder puncture soon after death. Compounds were identified from their mass spectra and Kovats retention indices. There were 53 possible endogenous scent compounds, 10 plant-derived compounds and 5 anthropogenic xenobiotics. Among the plant chemicals were several aromatic apocarotenoids previously found in greater abundance in the fox tail gland. They reflect the dietary consumption of carotenoids, essential for optimal health. One third of all the endogenous volatiles were sulfur compounds, a highly odiferous group which included thiols, methylsulfides and polysulfides. Five of the sulfur compounds (3-isopentenyl thiol, 1- and 2-phenylethyl methyl sulfide, octanethiol and benzyl methyl sulfide) have only been found in foxes, and four others (isopentyl methyl sulfide, 3-isopentenyl methyl sulfide, and 1- and 2-phenylethane thiol) only in some canid, mink and skunk species. This indicates that they are not normal mammalian metabolites and have evolved to serve a specific role. This role is for defence in musteloids and most likely for chemical communication in canids. The total production of sulfur compounds varied greatly between foxes (median 1.2, range 0.4–32.3 µg 'acetophenone equivalents'/mg creatinine) as did the relative abundance of different chemical types. The urinary scent chemistry may represent a highly evolved system of semiochemicals for communication between foxes.

## Introduction

The red fox (*Vulpes vulpes*, 'fox') is a highly successful animal that has a world-wide distribution and has adapted to a great variety of natural landscapes, from temperate and boreal forests, to deserts and arctic tundra, as well as agricultural and urban areas [1, 2]. Underlying this

**Data Availability Statement:** All relevant data are uploaded to a repository at the University of Tasmania (https://rdp.utas.edu.au) and publicly accessible via the following DOI: https://dx.doi.org/10.25959/sk1s-0081.

**Funding:** The authors received no specific funding for this work.

**Competing interests:** The authors have declared
that no competing interests exist.

success is a complex social structure [3, 4] that supports co-operative activity leading to a rate of population increase that can be very high under favourable conditions [5]. Foxes can flourish in proximity to humans, exploiting the abundance of shelter and food, and achieving a larger body size and population density than in the wild [2, 6, 7].

Increasing fox numbers have brought ecological, economic and health problems [1, 8]. Foxes eat a wide variety of food including insects, reptiles and other wildlife, farm and domestic animals, various fruits and other plant materials and food scraps [2, 9–15]. Their ecological impact is profound in Australia, where they have spread throughout the continent since their introduction in the 19th century [16]. They are a major cause of wildlife decline [1], and their great mobility leads to seed dispersal, often of pest species [17, 18]. Foxes are vectors for many zoonoses, including rabies, nematodes, and mites [19, 20]. When they become accustomed to humans, other nuisance behaviour such as bin-raiding and biting can occur [2].

Foxes are solitary foragers that can also live in groups [21, 22]; they travel long distances [18] and both defend [23] and share [4] territory. Studies in many other mammalian species have shown that communication between individuals relies on the production of odourant chemicals (semiochemicals) that can be detected by the olfactory system of conspecifics [24–27]. Semiochemicals are released in external secretions and excreta, and elicit responses related to the social, kinship, health, and reproductive status of the source animal.

The structure of fox society is also likely to depend on chemical signalling. Olfaction in most mammals involves two complementary systems, the main olfactory system and the vomeronasal system (VNS), which is particularly associated with reproductive behaviours and maternal recognition [28, 29]. Recently the anatomy and function of the fox VNS has been found to be well-developed and suited to assess the reception and recognition of semiochemicals [30]. In further evidence of their chemical signalling, foxes exhibit marking behaviour with urine [31–33] and faeces [34], and have an aromatic supracaudal tail gland [35] whose scent chemicals have recently been described [36].

It is over four decades since Jorgenson et al. [31] first identified scent chemicals in the urine of the red fox. Here we use the term "scent chemicals" for odourants whose signalling role has not been established. These authors noted earlier reports of apparent olfactory communication in foxes and other wild canid species and suggested that the compounds they found may serve this function.

Non-volatile compounds can also act as semiochemicals but require direct contact with the sensing animal to access the receptors of the VNS [37]. Volatile compounds, which act at a distance, are the focus of this study, which aims to update knowledge of fox urinary volatiles. The findings will inform future behavioural investigations into fox chemical communication. This in turn could enable the development of novel and improved methods of understanding and managing wild and urban fox populations.

## Materials and methods

### Animals and sample collection

Foxes are a declared pest species in Australia and are regularly culled by licensed hunters to protect livestock and native animals. The Animal Experimentation Ethics Committee of the University of Tasmania advised that there are no ethical issues in taking samples from animals that had already been legally killed by licensed hunters. Foxes were obtained during winter (July and August) from three farming (grazing and agriculture) regions in Victoria: the Anderson Peninsula, Skipton, and Koo Wee Rup.

Samples were taken as soon as possible (mostly within 30 min) after death, by syringe from the bladder after its exposure by an abdominal incision. The urine was immediately placed in a

glass vial with PTFE-sealed cap and kept on ice until frozen later that day. Afterwards, samples were stored at -80˚C until being thawed for analysis. The 15 foxes were adults, 13 males and 2 females, with no obvious indications of ill-health.

## Overview of analyses

The initial analyses used gas chromatography-mass spectrometry (GC-MS) and solid-phase microextraction (SPME) to identify volatile compounds in the headspace above fox urine. The relative abundance of each compound was determined as the fraction (%) of the total area of all measured peaks. These relative percentage values are interdependent, and it is more meaningful to measure the concentrations (eg as µg/ml urine) as these are independent of each other. Concentrations vary with water intake and urine flow, but can be standardised as the ratio to the concentration of creatinine, which enables meaningful comparisons to be made between individuals. This is the standard used for clinical urinary measurements [38].

This study was conceived as exploratory, and was not designed to quantitate the compounds found, which were mostly unknown. Quantitation of GC-MS analyses requires the addition of an internal standard to the urine sample before analysis. The finding that acetophenone was present in all urine samples, usually as the major constituent, indicated that its concentration could be determined and then used to estimate the concentration of other compounds in 'acetophenone equivalents'. This approximation ignores the expected differences between compounds in their extraction, chromatography and detection, but does avoid the problem of the interdependence of relative abundances.

Quantitation of acetophenone required another GC-MS analysis, with the addition of $d_8$-acetophenone as internal standard. These analyses showed that, after more prolonged storage, the amounts of some other compounds were reduced. Therefore, only acetophenone was analysed in the quantitative GC-MS analysis, and its concentration used with the peak area data from the first analysis to calculate the concentrations of other compounds in 'acetophenone equivalents'. Creatinine was determined by liquid chromatography-mass spectrometry (LC-MS), and the ratio of concentrations of compound/creatinine calculated.

This approach to quantitation was a consequence of the progressive development of our knowledge of the urinary compounds, and is neither conventional nor completely satisfactory, but does provide approximate data in biologically meaningful units. Given the provisional nature of the quantitative analysis, it has only been applied to the sulfur compounds which seem to be the most interesting group in the fox urine.

## Analysis by gas chromatography-mass spectrometry (GC-MS)

Solid-phase microextraction (SPME) used an automated system, Gerstel Multipurpose Robotic Sampler, (Gerstel GmbH, Mullheim, Germany) fitted to a Varian CP-3800 gas chromatograph coupled to a Brüker 300-MS triple quadrupole mass spectrometer. The SPME fibre was a 3-phase 50/30 µm DVB/CAR/PDMS Stableflex (Supelco Analytical) and was conditioned before use at 270˚C for 30 min in the GC injector.

Urine samples were thawed at room temperature, centrifuged and 0.5 ml placed in a 20 ml glass headspace tube closed with a PTFE-sealed magnetic screw cap. Kovats retention indices (KI) were determined by addition of a mixture of alkanes ($C_6$-$C_{27}$; Supelco Analytical) in 10 µl dichloromethane:hexane, 4:1. The concentration of acetophenone was determined by addition of an internal standard ($d_8$-acetophenone, Sigma-Aldrich), 300 ng in 10 µl ethanol.

The tubes were heated to 40˚C, and after 5 min pre-incubation the fibre was inserted into the headspace. The extraction proceeded for 30 min at 40˚C while the tube contents were mixed by repeated cycles of 400 rpm rotation for 5 s followed by 1 s rest.

Chromatography used an Agilent DB-5MS column, 30 m x 0.25 mm, with 0.25 μm phase thickness. The GC conditions were: injector 270˚C, He 1.2 ml/min, split 10:1 for 5 min then 20:1, oven 40˚C for 4 min, then 6˚C/min to 80˚C, 8˚C/min to 250˚C, 25˚C/min to 290˚C for 0 min (total time 33.52 min). Electron ionisation (EI) mass spectra were recorded in full scan mode using operating conditions described previously [39].

Samples were initially analyzed within 4 months of collection, and the acetophenone was quantitated subsequently. Some samples had been collected before the automated apparatus was available (foxes 1–8), and their initial analyses used manual extraction and injection methods previously described [40]. For all samples, acetophenone was quantitated using the automated method as described in **Overview of analyses**.

Four fox samples were also analyzed after extraction of urine with polydimethylsiloxane (PDMS) coated stir-bars (Twister®, Gerstel), following the method described by Zhang et al. [41]. This enabled some less volatile compounds to be identified, but they were not quantitated.

### Identification of compounds

Compounds were initially identified from matches of their mass spectrum (MS) and Kovats retention index (KI) with the searchable database in the NIST library [42]. Some compounds eluted before hexane, and their KI values were estimated by extending the KI-retention time plot with the early-eluting acetone and dichloromethane and using their literature KI values. Water blank samples showed several extraneous peaks, mainly Si compounds from the SPME fibre and, occasionally, traces of laboratory solvents; these were ignored.

Structures were confirmed by comparison with standards (obtained from Sigma Aldrich), when available, as well as by interpretation of their mass spectra where this gave characteristic ions. In particular, sulfur compounds were characterized by ions with the $^{34}$S isotope, two daltons heavier than the corresponding $^{32}$S ion and with about 4% of its abundance for each sulfur atom in the molecule [43]. Loss of the sulfur moiety (-SH, -SCH3, -CH$_2$SCH$_3$ radicals, and related whole molecules) produced ions that distinguished thiols from methyl sulfides, and the 1- and 2-thio-substituted phenylethane isomers from each other.

### Analysis by liquid chromatography-mass spectrometry (LC-MS)

Creatinine was determined in diluted (1:1000) urine by LC-MS, with the addition of d$_3$-creatinine (Santa Cruz Biotechnology, CA, USA), 5 ng/ml. The instrument, column and general operating conditions have been described previously [44]. The mobile phase was 0.1% v/v formic acid (98%): acetonitrile (2%) with flow rate 0.3 ml/min. The compounds eluted at 1.37 min. The mass spectrometer used positive electrospray ionization with multiple reaction monitoring, and the transitions for quantitation were $m/z$ 114.1→44.0 (creatinine) and m/z 117.1→47.0 (d$_3$-creatinine).

### Quantitation and statistical analysis

From the SPME-GC-MS chromatograms, compound abundances were determined by peak area, measured as total ion current (TIC), and expressed as the percentage of the total area of all peaks measured in each fox. In cases of co-elution, a good quality mass spectrum was found for each compound, from which up to three characteristic (and non-interfering) ions were selected. The total abundance of these quantitative ions (QI) in the mass spectrum was divided by the total of all ions (TIC) giving the ratio$_{QI/TIC}$. Thereafter, a selective chromatographic plot of quantitative ions gave a QI peak from which the expected TIC peak area was calculated (TIC$_{area}$ = QI$_{area}$/ratio$_{QI/TIC}$).

The GC-MS chromatograms showed that the TIC peaks for acetophenone and its $d_8$-isomer did not completely resolve. Their abundances were measured by ion plots of $m/z$ 120 + 105 + 77 for acetophenone (comprising 0.802 TIC) and $m/z$ 128 + 110 + 82 for $d_8$-acetophenone (0.783 TIC). The response factor acetophenone/$d_8$-isomer was 0.962.

The concentrations of sulfur compounds were estimated as 'acetophenone equivalents'. This was calculated from the ratio of their peak areas to that of acetophenone, whose concentration (μg/ml) was determined from the ratio of its peak area to that of the internal standard, $d_8$-acetophenone.

The relative amounts of compounds varied greatly, indicating that the data are best described by non-parametric statistics. The median was used as the most robust measure of the central value, and the maximum and minimum values to show the range. Compounds not found were considered as missing rather than zero values. The frequency of occurrence was expressed as the number of foxes in which the compound was found out of the total samples analyzed (N = 15 for SPME and 4 for stir bar extractions).

### Occurrence of compounds found in other mammals

We considered that the most interesting compounds, and the most likely to act as semiochemicals, would be those that are found most exclusively in foxes. Therefore, the literature was searched for reports of the presence of the fox volatiles in other mammals. The search used SciFinder (https://scifinder.cas.org/), with search terms including CAS no., scent, odourant, pheromone, urine, faeces, anal, gland, and secretion. The compounds were also searched for in the human metabolome database (https://hmdb.ca/) [45], whether in excreta, blood or normal metabolism. General information about sources and uses of chemicals was obtained from the Pubchem database (https://pubchem.ncbi.nlm.nih.gov/#).

## Results

Analysis by SPME-GC-MS produced a chromatogram with many peaks, illustrated in Fig 1 Examination showed 68 compounds of interest, and these are listed in Table 1 together with the internal standard (no. 31). Sixty compounds were found by SPME analysis and another 8 after stir-bar extraction. Fifty-three compounds were possible endogenous scent chemicals and the relative abundances of 47 were determined. There were also 10 plant-derived chemicals (abundances determined for 8) and 5 anthropogenic xenobiotics (abundances determined for 4). Ethyl acetate, although a known endogenous scent compound, is also a commonly used laboratory solvent making its quantitation unreliable without special precautions against contamination.

Table 1 summarizes the findings with compounds numbered and listed in order of elution by Kovats index, uniquely identified by CAS number, and assigned to informal chemical groups based on carbon skeleton and functionality. Fig 2 shows the number of foxes in which each compound was found. This varied from 1 to 15 (median 3), while the number of compounds found in individual foxes varied from 8–36 (median 19). Fig 3 summarizes the relative amounts of each compound found (as % total) as the quartiles (lower, 25%; median, 50%; upper, 75%) and range.

The results of the literature search are presented in Table 1 as findings in excreta and external glands in foxes, other canids, other mammals, and the human metabolome. Note that diet influences urinary composition, and many studies used captive animals whose diet may have differed significantly from that in the wild. The references for Table 1 are listed separately in the S1 Text.

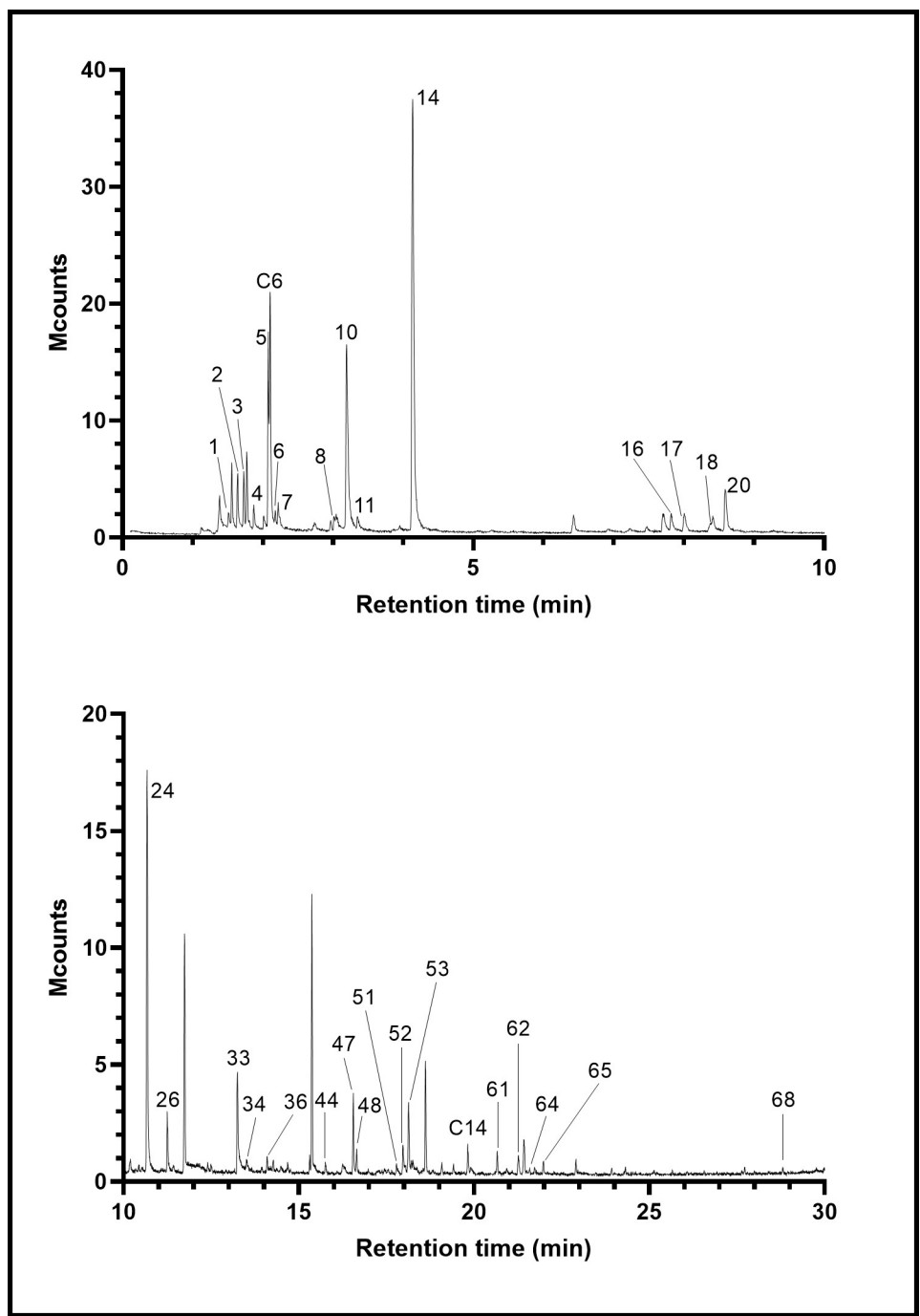

**Fig 1. SPME-GC-MS analysis of urine of a male fox (no. 6).** The peak numbers refer to compounds in Table 1, and C6 and C14 are hexane and tetradecane, respectively (both artifacts). Unlabelled peaks are fibre-related or other artifacts.

Fig 4 shows the relative amount of each compound found in individual foxes. Acetophenone (33) was the only volatile compound found in all foxes, and in most (13/15) it was the most abundant of the compounds whose peaks were measured (median 66.7, range 4.2–95.4%; Fig 3). It is commonly found in mammalian urine (Table 1). L-Phenylalanine is metabolized

**Table 1. Compounds found in fox urine and their reported findings in foxes, other canids and other mammals.**

| No. | Chemical name | KI | Rᵃ or KI lit. | CAS no. | Chemical groupᵇ | Fox | Wolf | Coyote Ref 10 | Dog | African wild dog Ref 14 | Black backed Jackal Ref 14 | Species, site, reference no. If more than 3, the no. of Spp. and site but no ref no. | Reports in HMDBᵈ |
|---|---|---|---|---|---|---|---|---|---|---|---|---|---|
| | Compound | | | | | | | Reports in canidsᶜ | | | | Reports in other mammalsᶜ | |
| 1 | Methanethiol | 474 | 473 | 74-93-1 | AS | | | | | | | Mouse U15, human U16 | N, U, F |
| 2 | 2-Propanone | 500 | R | 67-66-1 | K | | U6,7 | U | U11 A12 | | | Mouse U15, human U17 | N, U, F |
| 3 | Dimethylsulfide | 520 | R | 75-18-3 | AS | | U7 A8 | | A12 | | | 5 Spp. U | F |
| 4 | 1-Propanol | 551 | R | 71-23-8 | Alc | | U7 | | A12 | | | Mouse F18 | |
| 5 | 2-Butanone | 595 | R | 78-93-3 | K | | U6 | U | U11 A12 | | | 4 Spp. U | U, F |
| 6 | 2-Methyl-3-buten-2-ol | 607 | R | 115-18-4 | Alc | | | | | | | Mouse U19, African elephant U20 | |
| 7 | Ethyl acetateᵉ | 610 | R | 141-78-6 | E | T1 | | | A12 | | | Ubiquitous in eukaryotes | N, U, F |
| 8 | 2-Pentanone | 688 | R | 107-87-9 | K | | U6 | U | | | | 7 Spp. U | U, F |
| 9 | 3-Pentanone | 696 | R | 96-22-0 | K | | U6 | | | | | 4 Spp. U | |
| 10 | S-Methylthioacetate | 699 | R | 1534-08-3 | AS, E | | | | | | | Lion U21, mink A22 | F |
| 11 | 3-Hydroxy-2-butanone | 706 | 707 | 513-86-0 | K, Alc | | | | | | | Mouse U19, tree shrew U23, human U24 | U, F |
| 12 | 3-Isopentenyl alcohol | 729 | R | 763-32-6 | I, Alc | | | U | | | | Mouse U19, bobcat U25, deer U26 | F |
| 13 | 4-Methyl-2-pentanone | 730 | R | 108-10-1 | K | | U6 | | | | | Deer mouse U27, deer U26, African elephant U20 | U, F |
| 14 | Dimethyl disulfide | 742 | R | 624-92-0 | AS | | U7, A8 | U | U11, A12 | F | | 4 Spp. U, mink A22 | U, F |
| 15 | 3-Isopentenyl thiol | 794 | - | 58156-49-3 | IS | U2 | | | | | | | |
| 16 | 4-Heptanone | 871 | R | 123-19-3 | K | U2,3,4 | U6 | | | | | 7 Spp. U | U, F |
| 17 | Isopentyl methyl sulfide | 877 | R | 13286-90-3 | IS | U2,4 | U6,7 | U | U11 | | | | |
| 18 | Styrene | 888 | R | 100-42-5 | Ph | T1 | U6 F9 | | | U F | | 4 Spp. U, 2 Spp. F, 1 Sp. A, 3 Spp. G | F |
| 19 | 2-Heptanone | 889 | R | 110-43-0 | K | U2 | U6 A8 | U | | F | | 8 Spp. U | N, U, F |
| 20 | 3-Isopentenyl methyl sulfide | 896 | 883 | 5952-75-0 | IS | U2,3,4 | U6 | U | | | | Mink A22 | |
| 21 | 4-Butanolide | 907 | 908 | 96-48-0 | L | | | | | U F | | Lion U28, H U29, bat odour 30 | N, F |
| 22 | Dimethylsulfone | 914 | R | 67-71-0 | AS | | | | | U F | F | Mouse U15, cheetah U31 | N, U |
| 23 | Benzaldehyde | 952 | R | 100-52-7 | Ph, Ald | U3,4 T1 | U6 A8 F9 | | | U F A | F A | 6 Spp. U | U, F |
| 24 | Dimethyl trisulfide | 963 | R | 3658-80-8 | AS | | U7 A8 | U | | U F | | Human U16 | F |
| 25 | Phenol | 980 | R | 108-95-2 | PhOH | U2 | U6,7 A8 F9 | | | U F A | F A | 7 Spp. U, mouse F18 | U, F |

*(Continued)*

**Table 1.** (Continued)

| | Compound | | | | | Reports in canids[c] | | | | | | Reports in other mammals[c] | |
|---|---|---|---|---|---|---|---|---|---|---|---|---|---|
| No. | Chemical name | KI | Rᵃ or KI lit. | CAS no. | Chemical groupᵇ | Fox | Wolf | Coyote Ref 10 | Dog | African wild dog Ref 14 | Black backed Jackal Ref 14 | Species, site, reference no. If more than 3, the no. of Spp. and site but no ref no. | Reports in HMDBᵈ |
| 26 | 6-Methyl-5-hepten-2-one | 983 | R | 110-93-0 | I, K | U2,3 T1 | U6 | | | | | 6 Spp. U | F |
| 27 | β-Myrcene | 990 | 990 | 123-35-3 | Terp | T1 | | U | | | | Mouse U45 | F |
| 28 | Octanal | 999 | R | 124-13-0 | Ald | T1 | | U | | U F | F A | 7 Spp. U | F |
| 29 | Carbitol | 1008 | 1007 | 111-90-0 | AX, Alc, Ether | | | | | | | Exposed workers U32 | |
| 30 | 2,6,6-Trimethyl-cyclohexanoneᶠ | 1031 | R | 2408-37-9 | Apo, I, K | T1 | | | | | | | |
| 31 | d8-Acetophenone (internal standard) | 1055 | R | 19547-00-3 | AX, Ph, K | | | | | | | | |
| 32 | 1-Phenylethanol | 1056 | R | 98-85-1 | Ph, Alc | | | | | U F | | Lion U28, tree shrew U23, H exposed workers U33 | |
| 33 | Acetophenone | 1061 | R | 98-86-2 | Ph, K | U2,3,4 | U6,7 F9 | U | U47 | U F | | 11 Spp. U | F |
| 34 | 4-Methylphenol | 1071 | 1070 | 106-44-5 | PhOH | U2 | U7 F9 | | | U F | F | 7 Spp. U | U, F |
| 35 | Nonanal, branched | 1087 | - | | Ald | | | U | | U F | | | |
| 36 | Linalool | 1096 | R | 78-70-6 | Terp | U2 | | | | | | 4 Spp. U | U |
| 37 | Nonanal, n- | 1101 | R | 124-19-6 | Ald | T1 | | U | | U F | | 10 Spp. U, springbok G34 | N, U |
| 38 | 1-Phenylethane thiol | 1121 | R | 6263-65-6 | PhS | U2 | | | | | | Striped skunk A35 | |
| 39 | 1-Octanethiol | 1122 | R | 111-88-6 | AS | | | | | | | | |
| 40 | Benzyl methyl ketoneᶠ | 1125 | 1124 | 103-79-7 | Ph, K | | | | | | | 6 Spp. U | |
| 41 | Benzoic acid | 1154 | R | 65-85-0 | Ph | | U7 A8 | | U | U F A | F A | 4 Spp. U | N, U, F |
| 42 | Benzyl methyl sulfide | 1161 | 1167 | 766-92-7 | PhS | | | | | | | | |
| 43 | Ethyl benzoate | 1166 | R | 93-89-0 | Ph, E | U2 | | | | | | Hamster U36, deer G37, mandrill G38 | |
| 44 | 2-Phenylethane thiol | 1174 | R | 4410-99-5 | PhS | U2 | | | | | | Hooded & spotted skunk A39 | |
| 45 | 1-Benzothiophene | 1190 | R | 95-15-8 | AX, S | | | | | | | | |
| 46 | 1-Phenylethyl methyl sulfide | 1196 | - | 13125-70-7 | PhS | U2 | | | | | | | |
| 47 | Dimethyl tetrasulfide | 1215 | 1215 | 5756-24-1 | AS | | | | | | | Human U16 | |
| 48 | β-Cyclocitral | 1218 | R | 432-25-7 | Terp, Ald | U2 T1 | | | | | | | U |
| 49 | trans-Geraniolᶠ | 1254 | R | 106-24-1 | Terp, Alc | U2 | | | | | | 4 Spp. U | |
| 50 | Nonanoic acid, n- | 1260 | R | 112-05-0 | FA | T1 | | | | U | A | Tree shrew U23, lynx U40, tiger U41 | |

(Continued)

**Table 1.** (Continued)

| No. | Chemical name | KI | R[a] or KI lit. | CAS no. | Chemical group[b] | Fox | Wolf | Coyote Ref 10 | Dog | African wild dog Ref 14 | Black backed Jackal Ref 14 | Species, site, reference no. If more than 3, the no. of Spp. and site but no ref no. | Reports in HMDB[d] |
|---|---|---|---|---|---|---|---|---|---|---|---|---|---|
| | | | | | | | | | | | | **Reports in canids[c]** spanning Fox–Black backed Jackal; **Reports in other mammals[c]** | |
| 51 | 2-Phenylethyl methyl sulfide | 1280 | - | 5925-63-3 | PhS | U2,3,4 | | | | | | | |
| 52 | Indole | 1289 | R | 120-72-9 | Amine | U2 | U7 A8 F9 | | | U F | F A | 6 Spp. U | N, F |
| 53 | Unknown A m/z 41, 69, 123, 155 | 1299 | - | - | Unk | | | | | | | | |
| 54 | 2-Amino-acetophenone[f] | 1300 | 1299 | 551-93-9 | Ph, K, Amine | | | | | | | Ferret U A42, bat odour 30 | |
| 55 | 2-Methylquinoline[f] | 1309 | 1308 | 91-63-4 | Q | U2,3,4 | U7 | | U13 | | | 4 Skunk spp. A39, deer mouse U27, ferret U42 | |
| 56 | Ethyl hydrocinnamate | 1346 | 1347 | 2021-28-5 | Ph, E | | | | | | | Red deer ventral G U37 | |
| 57 | Decanoic acid, branched | 1348 | - | 334-48-5 | FA | | | | U | | | 4 Spp. U | F |
| 58 | Texanol isomer[f] | 1376 | 1380 | 74367-34-3 | AX, E, Alc | | | | | | | Mouse U34, human U33, tree shrew U23 | |
| 59 | Unknown B m/z 43, 69, 71, 135, 168 | 1404 | - | | Unk | | | | | | | | |
| 60 | 2-Methylquinoline, 8-hydroxy | 1443 | - | 826-81-3 | Q | | | | | | | | |
| 61 | trans-Geranyl acetone | 1451 | R | 3796-70-1 | Terp, K | U2,3 T1 | | | | | | 7 Spp. U, springbok G34, mouse G43 | |
| 62 | β-Ionone and its 5,6-epoxide | 1489 | R | 14901-07-6 | Apo, I, K | T1 | | | | | | Both in flying squirrel F44, β-ionone in human U24 | |
| 63 | Unknown C m/z 41, 69, 136, 155 | 1490 | - | | Unk | | | | | | | | |
| 64 | 2,4-Di-tert-butyl phenol | 1511 | R | 96-76-4 | AX, PhOH | | | | | | | 4 Spp. U | U, F |
| 65 | Dihydroactinidiolide | 1536 | 1538 | 17092-92-1 | Apo, I, L | T1,5 | | | | | | | |
| 66 | 4-Dodecanolide[f] | 1680 | 1681 | 2305-05-7 | L | | | | | | | Siberian hamster U36, tiger U41 | |
| 67 | 5-Dodecanolide[f] | 1711 | 1710 | 713-95-1 | L | | | | | | | Tiger U41 | |
| 68 | Octasulfur | 2057 | 2055 | 10544-50-0 | S | | F9 | | A12 | | | Cheetah U31, tiger U41, lynx U40, possum G46 | |
| 69 | Bisphenol A | 2173 | 2108 | 80-05-7 | AX, PhOH | T1 | | | | | | Mandrill G38 | N, U |

[a]Identification confirmed by a reference standard (R) or literature KI value (KI lit). *A dash (-) indicates that no literature KI value was found,*

[b]Chemical groups: Alc, alcohol; Ald, aldehyde; Amine; Apo, apocarotenoid; AS, alkyl sulfur compound; E, ester; FA, fatty acid; I, isoprenoid or norisoprenoid; IS, (nor) isoprenoid sulfur compound; K, ketone; L, lactone; Ph, phenyl compound; PhOH, phenol; PhS, phenyl sulfur compound; Q, quinolone compound; S, sulfur compound; Terp, terpenoid; Unk, unknown; AX, anthropogenic xenobiotic.

[c]Scientific species names (ICZN) may be found in the references cited. Sites where compounds were found: U, urine; F, faeces; A, anal sac or gland; T, tail gland; G, other glands; blank, not found.

Reference numbers are given after the site, or at the head of the column. *The references are listed in* S1 Text. *References for Table 1.*

[d]HMDB, Human Metabolome Database, *https://hmdb.ca/*, used here as a supplement to the reports in the general published literature. Additional abbreviation: N, normally found in human body or excreta (including xenobiotics from plants or commercial products).

[e]Ethyl acetate is ubiquitous in eukaryotes and a frequent urinary finding. It was not quantitated because it is also a commonly used laboratory solvent liable to contaminate SPME analyses.

[f]Only found in stir bar analyses, whose relative quantitative data could not be combined with those from SPME analyses.

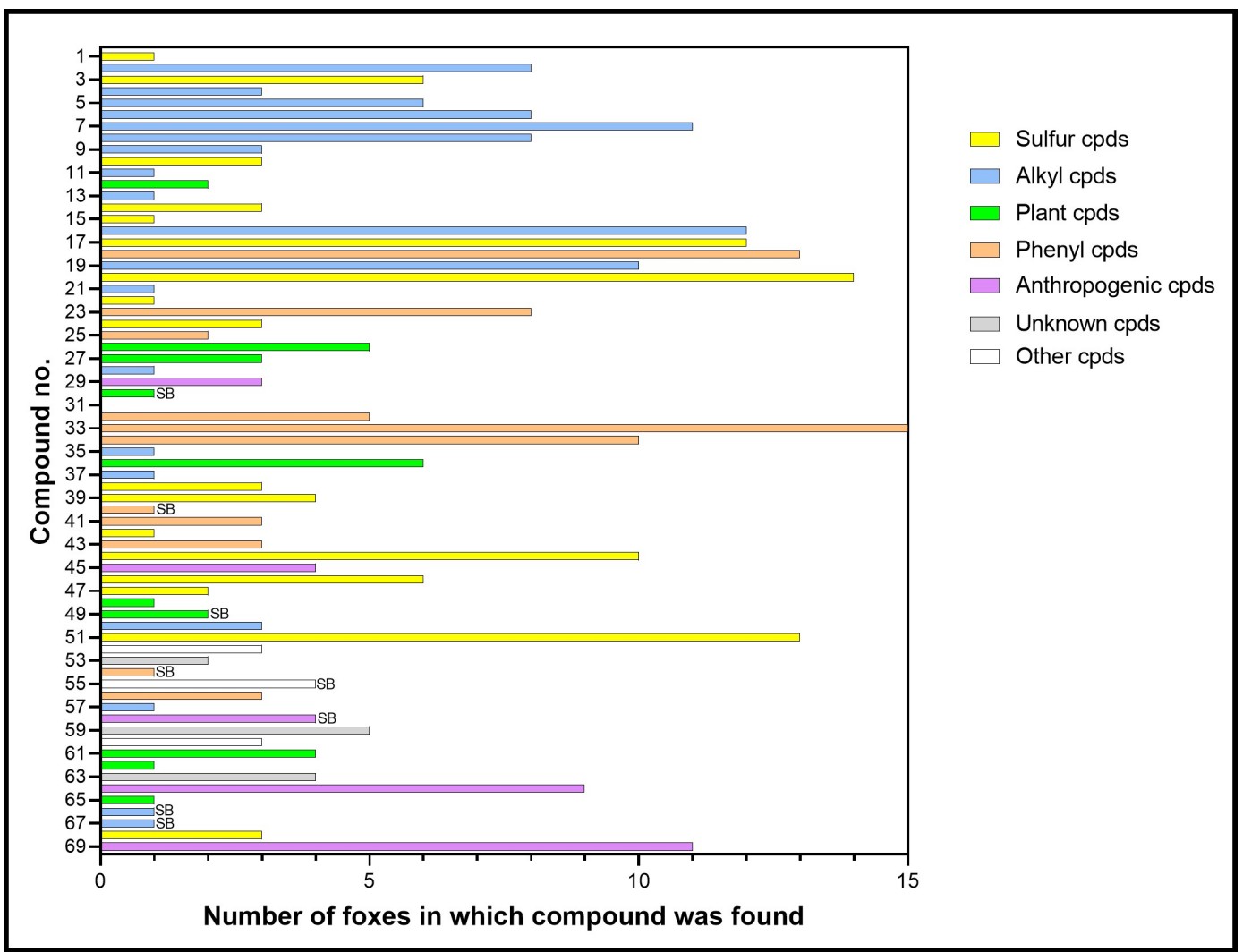

**Fig 2. Number of foxes in which each compound was found.** Only acetophenone (compound no. 33) was found in all 15 foxes. The internal standard (31) was only present when added, and 8 compounds were only found in stir bar (SB) analyses which were conducted on 4 samples. The colours indicate the chemical groups in Table 1.

to acetophenone and 1-phenylethanol (32) in plants [46] and diet may be the source of these compounds in foxes. The median acetophenone concentration was 6.0 (range 0.3–31.7) μg/ml. The fox (no. 6) in which acetophenone was least abundant (4.2%; Fig 4) had greatest relative amounts of three sulfur compounds: dimethyl disulfide (14, 33%), dimethyl trisulfide (24, 15%) and S-methylthioacetate (10, 13%). This illustrates the limitation in using relative percentages as a surrogate for quantitation: the values are interdependent. Unlike acetophenone, for nearly every other compound found the median was closer to the minimum than the maximum value, indicating that for most compounds there were only a few foxes in which there was a high relative abundance.

Sulfur compounds were the most significant findings in fox urine, accounting for 32% (17/53) of the endogenous scent compounds, and including the second most frequently found compound, 3-isopentenyl methyl sulfide (20), present in 14/15 foxes. Five sulfur compounds have only been found in foxes: 3-isopentenyl thiol (15), octanethiol (39), benzyl methyl sulfide

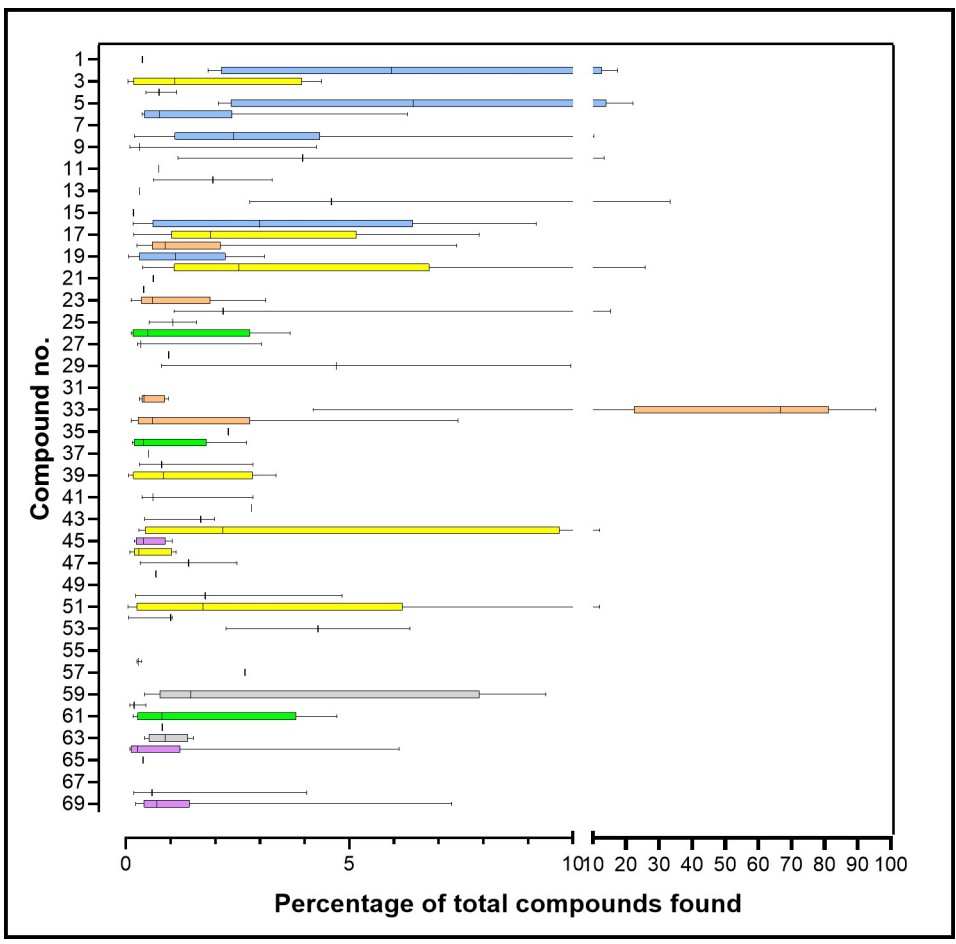

**Fig 3. The relative abundance (%total) of each compound in all foxes.** Boxes show the lower, median and upper quartiles, and the line gives the range of values. The number of foxes in which each compound was found is given in Fig 2. Eight compounds were only found in stir bar analyses and their amounts were not included. The colours indicate the chemical groups in Table 1 and Fig 2.

(42), and 1- and 2-phenylethyl methyl sulfide (46 and 51) (Table 1). Five others have only been reported in one or two other species: methanethiol (1, mouse and human), S-methylthioace-tate (10, lion and mink), 1-phenylethane thiol (38, striped skunk), 2-phenylethane thiol (44, hooded and spotted skunk) and dimethyl tetrasulfide (47, human). Isopentyl methyl sulfide (17) has only been found in canids, 3-isopentenyl methyl sulfide (20) in canids and mink, and dimethyltrisulfide (24) in canids and humans.

Elemental sulfur (68) has been reported in urine of several carnivores: canids (wolf and dog) and felids (cheetah, tiger and Eurasian lynx), and also in the paracloacal glands of the herbivorous brushtail possum. Unlike the other animals that excrete elemental sulfur, the felids did not excrete any sulfur compounds. The lactones have also been reported in felid urine: 4-butanolide (21) in the lion and 4- and 5-dodecanolide (66 and 67) in the tiger, which also excreted many other lactones [47].

The sulfur compounds are grouped in Table 1 according to similarities in their carbon skeleton or its absence: isoprenoid (IS), other alkyl (AS), phenyl (PS) and octasulfur (S). Each grouping may indicate a metabolic relatedness. For each fox, the relative amount of each group as a fraction of total sulfur compounds is shown in Fig 5. The individual variation in

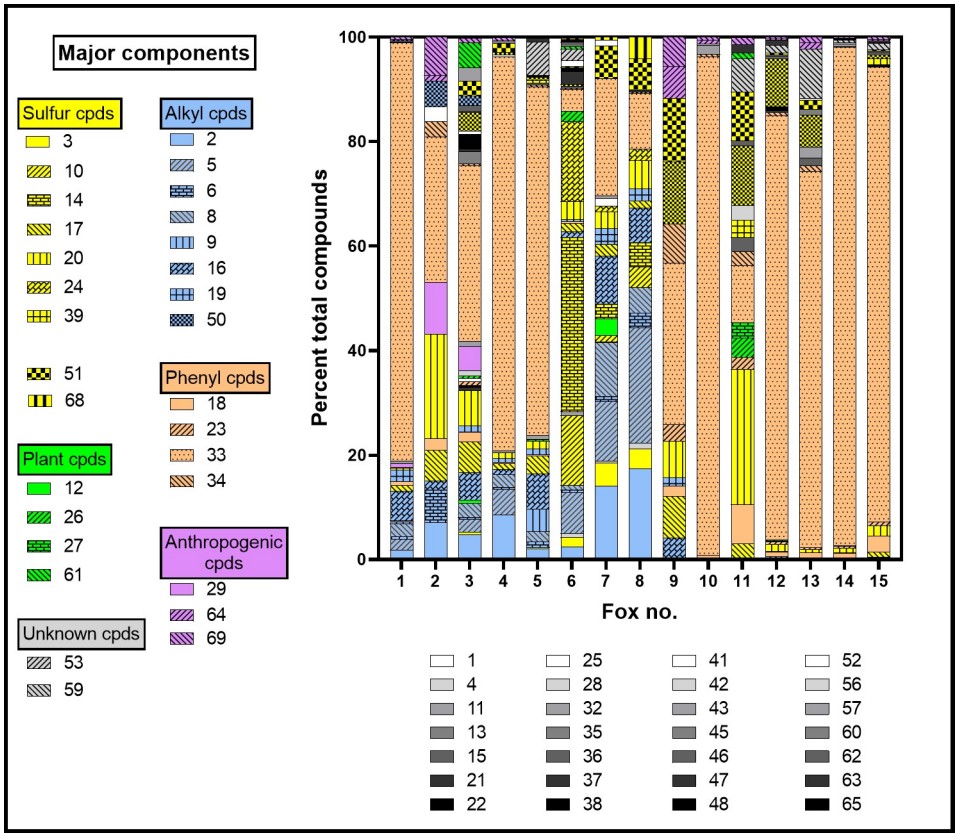

**Fig 4. The relative abundance (% total) of each compound found in individual foxes.** All data are plotted and compounds present at >3% in at least one fox have been individually coded as shown. The colours indicate the chemical groups in Table 1 and Figs 2 and 3. Foxes 12 and 13 were female.

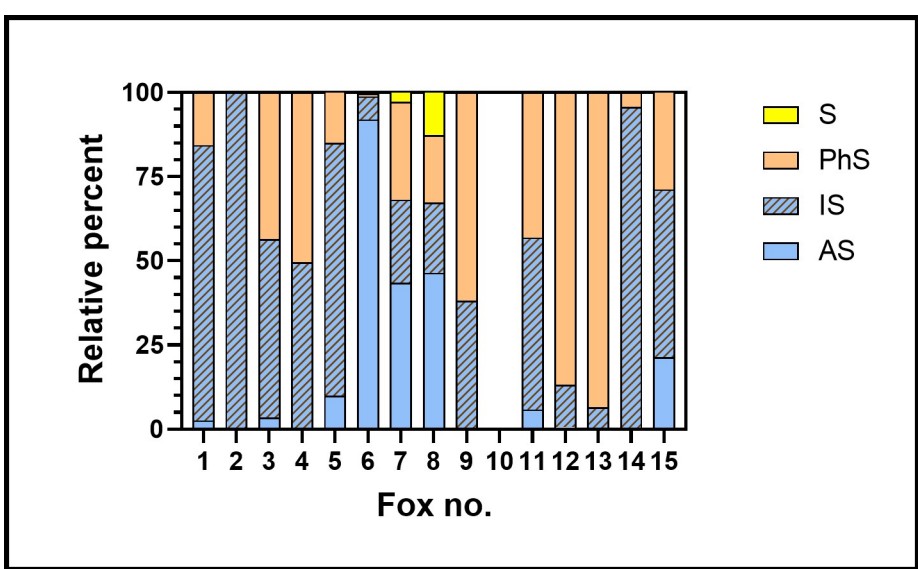

**Fig 5. Sulfur compounds found in the urine of 15 foxes.** Amounts are expressed as the percentage of all sulfur compounds in each fox, then grouped as in Table 1: alkyl (AS), isoprenoid (IS), phenyl (PhS) and elemental sulfur (S). Foxes 12 and 13 were female, and fox 10 produced no detectable sulfur compounds.

these proportions is striking, as is the absence of detectable sulfur compounds in one individual.

Urinary concentrations vary with the flux of water but can be adjusted to reflect the metabolic production of individual compounds. Creatinine is a protein degradation product that is produced and excreted in urine at an approximately constant rate, dependent on muscle mass but regardless of urine volume [48]. Urinary concentrations expressed as the ratio to creatinine give an amount that is independent of urine volume although it can be affected by body weight, age, or recent meals. This correction is routinely applied to normalize urinary concentrations of endogenous and xenobiotic compounds and enables meaningful comparisons between individuals. The median urinary concentration of creatinine in foxes was 0.73 (range 0.32–2.85) mg/ml, similar to that reported in dogs [48]. The corrected urinary concentration of acetophenone, the most abundant compound, was: median 4.6 (range 0.3–30.7) μg/mg creatinine.

The total concentration of endogenous sulfur compounds in each animal was estimated from the ratio of their total peak areas to that of acetophenone. This is only an approximation, as extraction and response factors differ between compounds. The total urinary concentration of sulfur compounds varied greatly between foxes: median 1.1 (range 0.2–11.2) μg/ml in acetophenone equivalents. Division by creatinine concentration gave amounts that varied even more, and showed that the metabolic production of sulfur compounds varied greatly between foxes, even when fox no. 10 (in which none was detectable) was excluded: median 1.2 (range 0.4–32.3) μg 'acetophenone equivalents'/mg creatinine. Fox 10 had the lowest urinary creatinine concentration (0.32 mg/ml) but its acetophenone concentration was 8.0 μg/ml, higher than the median for all foxes (6.0 μg/ml). After correction, its concentration per mg creatinine was still high (25.3 μg acetophenone/mg creatinine).

Another chemical theme was the presence in fox urine of many isoprenoids and related compounds that are well-known as plant chemicals. Several have also been found in the fox tail gland: 2,6,6-trimethylcyclohexanone (30), β-cyclocitral (48), β-ionone and its 5,6-epoxide (which co-eluted, 62) and dihydroactinidiolide (65) [36]. Other plant chemicals are commonly occurring urinary metabolites: 3-isopentenyl alcohol (12), *trans*-geraniol (49), *trans*-geranylacetone (61), and the terpenes β-myrcene (27) and linalool (36).

2-Aminoacetophenone (54) has only previously been found in ferret urine and faeces [41], and in the body odour of the Mexican bat, *Tadarida brasiliensis* [49]. 2-Methylquinoline (55) is found in anal sacs of 4 species of skunk [50] and the urine of another musteloid, the ferret [41], as well as urine of fox, wolf and dog (Table 1). Its 8-hydroxy-derivative (60) has only been found in the fox and may be an oxidized metabolite.

There were five compounds of human origin that may have been present in foxes due to their ingestion of contaminated plants, food scraps or ground water.

Carbitol (29), 2-(2-ethoxyethoxy)ethanol, is a synthetic chemical used in many products including pesticides and cleaners. It was found in only 3 foxes and has been reported in the urine of exposed workers [51], but not previously in wild animals.

Texanol B (58) is a monoester of isobutyric acid and 2,2,4-trimethyl-1,3-pentanediol, mostly used as an undefined mixture of the two isomers. Annual production in the USA is 45–113 thousand metric tons, and it has multiple uses in domestic and industrial products, including paints, adhesives and coatings. It has been found in waste water from agricultural [52] and household sources [53]. It was present in all four stir bar fox urine analyses, and has been found in urine of humans and captive animals but not in wildlife, presumably due to their lack of exposure.

2,4-Di-tertiary-butylphenol (64, 2,4-DTBP) is an antioxidant used in foods and plastics from which it leaches out into the environment [54]. It was present in 9/15 fox urines and has

been found at high levels in human urine; unexpectedly more than its 2,6-isomer, also a widely-used antioxidant, that has been the isomer of most concern [55]. In captive mice, separate urinary studies have found either the 2,4-isomer [56] or the 2,6-isomer [57]. The latter was also found in the urine of captive lions [58] and tree shrews [59] and the sternal gland of mandrills [60].

Bisphenol A (69, 4-[2-(4-hydroxyphenyl)propan-2-yl]phenol; BPA) is used in many products, including polycarbonate and other plastics (importantly, food packagings), building materials and paper coatings [61]. Annual production in 2011 was 5.5 million metric tons, and it leaches into the environment, where it now has a world-wide distribution in water and effluents [62]. It was present in most fox urine and was previously found in their tail gland [36]; it is also frequently found in humans and aquatic animals but has not been reported in wild terrestrial mammals. There is one report of BPA in the sternal gland of the mandrill, held in captivity [60].

1-Benzothiophene (45) is an environmental pollutant from petroleum [63] but it has also been found to be a plant defence chemical, being produced by maize (*Zea mays*) in response to fungal infections [64]. It has not been previously reported in mammals, although has been found in farm-raised catfish [65]. We consider that the fox urinary benzothiophene is exogenous.

The remaining compounds found were mostly frequent occurrences in mammalian urine, faeces or glandular secretions, although this does not preclude them from being fox semiochemicals. A few have been characterized as mammalian signalling compounds: 4- and 2-heptanone (16 and 19), acetophenone (33), 4-methylphenol (34), benzoic acid (41) and 4-dodecanolide (66) [26].

There were three compounds that could not be identified although they were significant findings in several foxes (Figs 2 and 3). Unknown A (53) had a KI of 1299 and showed ions at *m/z* 155.2 (80%), 123.2 (21%), 101.2 (13.2%), 68.2 (15%), 69.2 (100%), 67.2 (26%) and 41.1 (44%). Unknown B (59) had a KI of 1404 and showed ions at *m/z* 168.2 (21%), 135.2 (43%), 109.2 (47%), 82.2 (46%), 71.2 (100%), 69.2 (81%) and 43.1 (40%). Unknown C (63) had a KI of 1490 and showed ions at *m/z* 155.2 (17%), 136.3 (25%) and 69.2 (100%).

## Discussion

This study has extended our knowledge of the volatile scent compounds in fox urine and has indicated areas for further investigation. Although urine and its constituents have a well-established role in chemical signalling between conspecifics [24], the primary role of urine production is for metabolic homeostasis, by excretion of waste products and maintenance of normal body salt and water levels [38]. Wastes include the end products of metabolism (eg creatinine from amino acids), surpluses of salts and water, and xenobiotics without nutritional value, such as plant secondary metabolites and anthropogenic synthetic chemicals. Animals learn to associate particular odours with characteristics of the emitting animal, for example detecting dietary-related compounds [66] or microbial products in a conspecific [67] or identifying predator or prey species [68]. Pheromones seem to have arisen when an odourant compound emitted by one animal became a cue to a property of significance to a receiving conspecific, followed by co-evolution of the chemical structure and olfactory receptors to produce an innate system that is both sensitive and specific [24, 69, 70].

Thus responses to signalling compounds can be learned or innate, and in the absence of behavioural studies it is not possible to exclude a signalling role for any of the compounds found in fox urine. Indeed, it may be a combination of chemicals that conveys a signal [24] although it has been argued that a single compound is more likely [71, 72]. Considering the

many different responses that are associated with semiochemicals [26, 73] it seems likely that a number of chemical constituents are involved.

Mammals share a common basic metabolism and resultant urinary composition, so it is not surprising that many of the volatile compounds found in fox urine are also present in other mammals. However, there are several compounds found only in foxes or in few other mammals, indicating that they are not normal mammalian metabolites and that their production has evolved to serve a functional role in foxes.

In particular, a third of fox urinary volatiles are sulfur compounds, a notoriously odiferous group, with a limited distribution amongst other animals. Three different carbon groups (alkyl, isopentyl and phenylethane) formed the sulfur compounds indicating distinct metabolic pathways in their synthesis. Once formed as thiols, subsequent methylation can readily produce the corresponding sulfide. Pheromone diversity can evolve from such small chemical changes [74].

The phenylethane group is present in 1- and 2-phenylethane thiol, which are only otherwise found in three skunk species where they are produced in secretory glands that line the anal sacs [50, 75]. Skunks use the foul-smelling sac contents as a defensive spray.

The corresponding methyl sulfides of 1- and 2- phenylethane thiol have only been found in fox urine. It is not known where the four phenylethane sulfur compounds are produced in the fox, whether the liver or kidney or some unknown gland. Lions and other big cats produce a marking fluid containing many scent compounds that can be voided with urine [76, 77], but this has not been investigated in canids.

Isopentyl methyl sulfide and its unsaturated cogener, 3-isopentenyl methyl sulfide, share the same branched 5-carbon group and have only been found in the urine of three other canids and the anal sac of the mink, a musteloid related to skunks [78]. Minks can expel the contents of their anal sacs and use this for defence and to mark territory. The corresponding 3-isopentenyl thiol has only been found in fox urine, while the related 3-isopentenyl alcohol is found in several other mammals.

S-Thiomethyl acetate has also been found in mink anal sacs [78], and in the lipid-based marking fluid which is voided with lion urine [58]. It is formed by acetylation of methanethiol. It was found in the largest proportion in the one fox in which the precursor methanethiol was also present.

Skunks are sparing in their use of their defensive spray, as it is only slowly replenished [79, 80]. This is likely to be due to a limited supply of metabolic precursors. Cysteine and other sulfur-containing amino acids (SAAs) are precursors of glutathione and other defence chemicals [81, 82]. Production of glutathione is limited by SAA availability and plant proteins tend to be deficient in SAAs [83]. Animal diets are a better source of sulfur compounds [84].

All but one of the fox urine samples contained a suite of sulfur compounds, although the total amount was highly variable. Dietary-related compounds can represent a reliable or "honest" signal of good nutritional status, which can influence attractiveness and mate choice [66]. If sulfur compounds are required for a high quality diet, then their abundance in urine could indicate just such an honest signal to foxes.

The acetophenone finding may be quite significant, since in no other animal has it been found in such abundance. Acetophenone has been found in all previous fox urine studies but only sometimes found in the urine of other species, suggesting that its production is variable. It has been found in other canids (Table 1), and in wolf urine it comprised 1–2% total volatiles [85]. The excretion of acetophenone was sex-dependent in wolves, being greater in females and castrated males [86, 87]. Recently acetophenone has been reported in dog urine where it increased during oestrous and apparently acts as a semiochemical with complex effects on mating behavior [88, 89]. Although acetophenone is not usually reported in human urine, in

one study its excretion was found to increase after the experimental induction of inflammation [90].

Most of the urinary isoprenoids are known plant products and could originate in the fox's diet that, as discussed above, can have a significant plant content. Isoprenoids are synthesized in plants from Δ3-isopentenyl diphosphate, and include terpenoids, carotenoids and steroids [91]. Mammals synthesize steroids the same way, but only plants can produce terpenoids and carotenoids. However, mammals can synthesize the C5-carbon skeleton required for the three sulfur-containing isoprenoids found in fox urine: 3-isopentenyl thiol, 3-isopentenyl methyl sulfide and isopentyl methyl sulfide, as well as 3-isopentenyl alcohol.

Terpenoids are produced from an early branch of the isoprenoid chain development [92]. The terpenoids, β-myrcene, linalool, β-cyclocitral, *trans*-geraniol and *trans*-geranyl acetone, are volatile C10-13 scent compounds present in leaves and flowers [93], and are frequent urinary findings in herbivores and omnivores. 6-Methyl-5-hepten-2-one (sulcatone) is a terpenoid commonly found in animal urine and human body odour where it acts as an attractant to mosquitoes that have adapted to human prey [94]; it is also common in insect secretions [95, 96].

Carotenoids are C40 plant pigments that colour fruits and flowers [97, 98]. Their breakdown products, apocarotenoids, are plant signalling and defence molecules [99, 100]. Low molecular weight volatile apocarotenoids are potent odourants that attract pollinating insects, and include β-ionone, β-ionone-5,6-epoxide, 2,4,6-trimethylcyclohexanone and dihydroactinidiolide [98, 101]. 6-Methyl-5-hepten-2-one and β-cyclocitral can also be formed by carotenoid degradation [98, 101].

These and other apocarotenoids were also found in the fox tail gland secretions [36], but are not known in other mammals. Animals require carotenoids and apocarotenoids for vision (eg retinoids) and reproductive and general health, and obtain them by eating plants or, for carnivores, eating herbivores [101, 102]. Thus, their expression as odourant molecules may act as a reliable olfactory signal of good nutrition. Many birds advertise the quality of their carotenoid consumption by using the pigments to colour their feathers or skin, and this enhances their attractiveness for mating [102–104]. In foxes, considering their nocturnal and covert lifestyle and well-developed VNS, an olfactory signal would be more effective than a visual one.

The absence of these compounds in other canids may reflect the difference in their diets. Foxes eat a significant amount of plant material [10], especially fruits [17, 105], which are a good source of carotenoids [102, 106]. The grey wolf, *Canis lupus*, is mainly carnivorous in the wild with a small proportion (5–10%) of fruit in its diet [107]. However, detailed comparisons cannot be made, as although the fox studies cited in Table 1 were based on wild animals, only one of the wolf studies [108] used wild animals, and all the others as well as the coyote and domestic dog studies used captive animals fed an artificial diet whose composition is not determined by animal choice.

The finding of several anthropogenic chemicals in fox urine shows that foxes, although living in the wild state, are not free from human impacts when they forage around farms and human habitation, especially as their diet is omnivorous. Similarly, the other reported findings of texanol B, 2,4-DTBP and BPA were in captive animals rather than those living freely away from human activities (Table 1 references, S1 Text).

BPA and the other phenolic, 2,4-DTBP, are mostly excreted as their glucuronide conjugates [55, 62] that would not be detected by our analysis of urinary volatiles. This indicates that the mostly low levels found in fox urine were an underestimate of the total amount. It is also possible that our findings were artifactual, given the prevalence (especially of BPA) in the environment. However, the samples were stored in glass/PTFE containers and only briefly exposed to plastic in the urine collection syringe; also, BPA and DTBP were not present in every fox

sample, and the highest levels (7 and 6% total) were quite significant; therefore we consider that these compounds were present in fox urine.

Several urinary compounds are synthetic commercial chemicals that are also natural products and sometimes found as environmental contaminants, so they have several possible sources, not necessarily mutually exclusive. 2-Aminoacetophenone is a plant product and can also be produced endogenously in mammals from tryptophan via kynurenine [109]. 2-Methylquinoline is an environmental pollutant present in petroleum and many commercial products [110, 111]. It has been found in several animals, notably canids and 5 musteloid species, and it may also have endogenous sources. Compound 60, 8-hydroxy-2-methylquinoline, could be formed by oxidation of 2-methylquinoline but it is also a widely-used compound itself which has been found in landfill leachate [112]. Texanol is another synthetic compound that may be also produced naturally, as it has been reported in the plant metabolome [113] and in volatiles from Italian cherry plums [114].

In most cases the effects of anthropogenic xenobiotics on wildlife are unknown and, except for potent toxins with obvious effects, in practice unknowable. There have been many studies of compounds of low toxicity and high use but their actual risk to wildlife remains uncertain. For example, BPA is of concern because of its prevalence and potential endocrine, mutagenic and carcinogenic effects; nevertheless its toxicity to humans at current levels of exposure remains uncertain, and data on mammalian wildlife are lacking [61, 62, 115].

## Conclusions

Sixteen sulfur compounds were detected in fox urine: 5 have only been found in foxes and 4 others only in some canids and musteloids. Given that this exclusivity shows they are not usual mammalian metabolites and considering that there is a cost in producing any metabolites, it is proposed that they have a beneficial function, most likely as semiochemicals for communication between foxes. Notably, both the total amount of sulfur compounds produced, and their relative proportions, varied greatly between individuals. Based on these findings, we hypothesize that this represents a highly evolved system of semiochemicals for communication within fox communities. From studies with other mammalian species, the possible signalling is likely to include information on an individual's identity and its nutritional, health, social and reproductive status.

The plant metabolites found in urine indicate the importance of plants in the fox diet. Together with their presence in the tail gland, they may also contribute to chemical signalling. In particular, the apocarotenoids signal a significant consumption of plant carotenoids, essential for mammalian health, possibly acting as an honest sign of good nutrition, as has been observed in bird species.

Taken together, the abundance of scent compounds suggests that future behavioural investigations may be able to exploit a semiochemical approach to better manage fox populations in the wild.

The presence of several anthropogenic chemicals in fox urine is further evidence of their pervasiveness in the environment and, given their actual and potential toxicity, the need to monitor their levels and control their release.

## Supporting information

**S1 Text. References for Table 1.**
(DOCX)

**S2 Text. Identification of compounds without a reference KI value.**
(DOCX)

**S1 Fig.**
(TIF)

## Acknowledgments

We are grateful to the many people who assisted and made this study possible. Stuart Murphy and Duncan Sutherland, Phillip Island Nature Parks, and members of the Anderson Peninsula Fox Hunting Club assisted in obtaining fox samples in eastern Victoria. Charlie Robinson, Mooramong Nature Reserve, Shane Grose, and Glenn Gray assisted in obtaining fox samples in the Skipton area, western Victoria. Jack West and the Koo Wee Rup fox hunting group assisted in obtaining fox samples south-east of Melbourne. All contributed helpful discussions on foxes, as did Clive Marks and Lyndall Rowley. David Obendorf participated in the field work and the discussions on foxes, their biology and behaviour. Patricia Fleming, Murdoch University, assisted with data on fox plant consumption. The fox image was supplied by Sean Passarin, Wildlife Photographer. We thank the anonymous reviewers for their helpful comments which improved the manuscript.

## Author Contributions

**Conceptualization:** Stuart McLean, Noel W. Davies.

**Data curation:** Stuart McLean, David S. Nichols.

**Formal analysis:** Stuart McLean, David S. Nichols, Noel W. Davies.

**Investigation:** Stuart McLean, David S. Nichols, Noel W. Davies.

**Methodology:** Stuart McLean, David S. Nichols, Noel W. Davies.

**Project administration:** Stuart McLean.

**Resources:** Stuart McLean, David S. Nichols, Noel W. Davies.

**Supervision:** Stuart McLean, Noel W. Davies.

**Validation:** Stuart McLean, David S. Nichols.

**Writing – original draft:** Stuart McLean.

**Writing – review & editing:** Stuart McLean, David S. Nichols, Noel W. Davies.

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
