## [Decision Letter · Decision Letter 0]

30 Dec 2020

PONE-D-20-33017

Volatile Scent Chemicals in the Urine of the Red Fox, Vulpes vulpes

PLOS ONE

Dear Dr. McLean,

Thank you for submitting your manuscript to PLOS ONE. After careful consideration, we feel that it has merit but does not fully meet PLOS ONE’s publication criteria as it currently stands. Therefore, we invite you to submit a revised version of the manuscript that addresses the points raised during the review process.

We look forward to receiving your revised manuscript.

Kind regards,

Bi-Song Yue, Ph.D

Academic Editor

PLOS ONE

Reviewers' comments:

Reviewer's Responses to Questions

**Comments to the Author**

1. Is the manuscript technically sound, and do the data support the conclusions?

Reviewer #1: Yes

Reviewer #2: Partly

2. Has the statistical analysis been performed appropriately and rigorously? 

Reviewer #1: Yes

Reviewer #2: No

3. Have the authors made all data underlying the findings in their manuscript fully available?

Reviewer #1: Yes

Reviewer #2: No

4. Is the manuscript presented in an intelligible fashion and written in standard English?

Reviewer #1: Yes

Reviewer #2: Yes

5. Review Comments to the Author

Reviewer #1: Strengths

One strength of this work is it has revealed a greater number of compounds in the urine of red foxes than previous studies. As mentioned by the authors, the urine of red foxes was first analyzed four decades prior to this work by Jorgenson et al. [1] who used gas chromatography with a nitrogen and sulfur selective detectors, which only allowed for the identification of eight compounds. In contrast, the authors of this work used gas chromatography-mass spectrometry (a universal detector), which has allowed them to detect and presumptively identify 69 compounds from red foxes. By revealing a larger array of compounds, scientists can now begin to explore their role in chemical signaling among conspecifics.

Another strength to this work is the sampling methods. By removing urine directly from the bladders of culled red foxes (often within 30 minutes of death), they were able to retrieve a native sample that is free of any external materials. This ensured that their results were unambiguous.

Lastly, I appreciated how the authors tried to accurately identify the compounds. It is common practice to use NIST library matches to get a tentative idea of which compounds are present in samples, but I believe it is essential to use chemical standards to confirm their identity. However, I understand it can be difficult to obtain a standard for each compound when working with upwards of 50 compounds. I was impressed to see they were able to purchase standards for 41 compounds and then use Kovat’s retention index to confirm the rest. These combined efforts allowed them to positively identify a total 59 compounds.

Weaknesses

The data was a bit overwhelming to analyze and isn’t user friendly. The primary source of data was presented in Table 1, which was chock-full of information thus making it hard to process. In my opinion, it would be best to take the information in Table 1 and distribute it into different tables and graphs, each with their own objective. For instance, a bar graph could compare the number of foxes a compound was identified from. Another figure could include box and whisker plots of the percentage of total compounds quantitated for each compound. Lastly, a table could be used to compare which compounds were identified in other species. Additionally, I was disappointed to not see information specifically pertaining to the foxes sampled. It would have been interesting to see which compounds and the amounts of each compound (according to peak area) was collected from each fox in a stacked column graph.

On occasion, I found some of the descriptions hard to understand. The language was unambiguous, but the ideas were not presented in a clear and logical way. For example, the paragraph on page 6, lines 124-129 (under the section titled “Analysis by gas chromatography-mass spectrometry”) was not clear to me (see my comments in minor issues). I felt the same way about the “Quantitation and Statistical Analysis” section. I think it is safe to say that quantifying compounds was not the main objective of this paper; however, an attempt was made to quantify acetophenone and the sulfur compounds and I found these methods to be unconventional. There were a few more instances of confusing text in the results and discussion sections, as well.

Minor Issues

• Analysis by gas chromatography-mass spectrometry, lines 108-116: It is unclear when the deuterated acetophenone and the alkane mixture was added to the urine sample. These additives are mentioned after compound extraction. Please clarify.

• Analysis by gas chromatography-mass spectrometry, lines 124-129: This paragraph is confusing. Is this saying that some samples were analyzed using automated injection (robotic sampler) and the rest were analyzed using manual injection? Also, what was meant by the statement acetophenone was quantitated after samples were analyzed (line 124-125)? Please rewrite in a clear manner.

• Quantitation and Statistical Analysis: The word ‘quantitation’ is a bit misleading since mass amounts were not calculated and should probably be referred to as ‘compound abundance’ instead. Consider revising.

• Quantitation and Statistical Analysis: In terms of acetophenone, the methods do not clearly state how it was quantified. If the concentration of acetophenone was calculated by comparison to deuterated acetophenone, then it is important that a response factor be established between the two compounds. Add text that explains how acetophenone was quantified.

• Figure 1: Is deuterated acetophenone included in this sample? If so, I think it would be best to indicate its peak.

• Table 1: As I mentioned previously, I think it would be best to distribute the information in Table 1 into multiple different tables and graphs, each with their own objective. For instance, a bar graph could compare the number of foxes (y-axis) a compound was identified from (x-axis). Another figure could include box and whisker plots of the percentage of total compounds quantitated for each compound. Lastly, a table could be used to compare which compounds were identified in other species.

• Table 1, RI or KI lit: Some compounds have a dash listed. How were their identities confirmed?

• Table 1, Reports in canids: The references do not appear to be correct. For example, with regards to ethyl acetate, T1 is listed under fox. I interpret that as ethyl acetate was detected in the tail gland of a fox as it appears in reference #1, but reference number 1 by Saunders et al. [2] does not mention ethyl acetate anywhere in their review. Update references in the table to match the correct citation.

• Table 1, Human HMDB: Under beta-myrcene it lists, ‘check’. Is that supposed to be there?

• Table 1, Human HMDB: An X indicates that the compound was not reported in the database. I would suggest this be removed to be consistent with the other species. This table is already full of information, it might be best to remove unnecessary symbols.

• Table 1, Human HMBD: The symbols are confusing. Only N and X have been described, but symbol such as U and F are included. Does U mean urine and F mean feces as seen in the other columns?

• Results, Line 304: It is mentioned that 2-aminoacetophenone was previously found in ferret urine and feces, but no reference is provided. Add reference.

• Results, Line 310: The sentence begins by saying six compounds of human origin were present in fox urine, which includes deuterated acetophenone; however, wasn’t deuterated acetophenone added to each sample? Consider revising this sentence.

References

1. Jorgenson J, Novotny M, Carmack M, Copland G, Wilson S, Katona S, et al. Chemical scent constituents in the urine of the red fox (Vulpes vulpes L.) during the winter season. Science. 1978;199(4330):796-8.

2. Saunders GR, Gentle MN, Dickman CR. The impacts and management of foxes Vulpes vulpes in Australia. Mammal Review. 2010;40(3):181-211.

Reviewer #2: In this manuscript entitled “Volatile scent chemicals in the urine of the red fox, Vulpes vulpes” Stuart Mc Lean et. al. studied the chemical composition of fox urine using solid-phase microextraction and gas chromatography-mass spectrometry to analyze the different urinary volatiles in a number of free-ranging wild foxes living in farmlands and bushes in Victoria, Australia. Compounds were identified from their mass spectra and Kovats retention indices. Identified compounds include endogenous scent compounds, various plant derived compounds and anthropogenic xenobiotics. These urinary odorant compounds may represent a highly evolved system of semiochemicals for communication.

The results depicted are well presented, correlated to each other and properly explained (except in some aspect) providing a deep insight into the semiochemical interaction. The work depits a very interesting behavioural aspect of the species, thus apt for the theme of this journal and the manuscript may be accepted for publication once these points are addressed:

1.The results demand the raw data of the gas chromatography-mass spectrometry analysis, may be presented as supplimentary file.

2.The percentage of the compounds should be represented by statistical analysis with ± values.

3.In the text the authors mentioned the urinary volatiles from both male and female species. How the composition of the chemicals varied in male and female species? Was there any major differences in the composition? What is the significance of such variations in the volatile composition? It will be appreciated if the authors highlight these aspects with proper explanation.

4.Minor errors: Take care of the few typographical errors in the text.

6. PLOS authors have the option to publish the peer review history of their article (what does this mean?). If published, this will include your full peer review and any attached files.

Reviewer #1: No

Reviewer #2: No

---

## [Author Response · Author response to Decision Letter 0]

25 Feb 2021

All comments have been responded to in the "Response to reviewers" file.

---

## [Editor Report · Decision Letter 1]

9 Mar 2021

Volatile Scent Chemicals in the Urine of the Red Fox, Vulpes vulpes

PONE-D-20-33017R1

Dear Dr. McLean,

We’re pleased to inform you that your manuscript has been judged scientifically suitable for publication and will be formally accepted for publication once it meets all outstanding technical requirements.

Kind regards,

Bi-Song Yue, Ph.D

Academic Editor

PLOS ONE

---

## [Editor Report · Acceptance letter]

22 Mar 2021

PONE-D-20-33017R1 

Volatile Scent Chemicals in the Urine of the Red Fox, *Vulpes vulpes*

Dear Dr. McLean:

I'm pleased to inform you that your manuscript has been deemed suitable for publication in PLOS ONE. Congratulations! Your manuscript is now with our production department. 

Kind regards, 

on behalf of

Dr. Bi-Song Yue 

Academic Editor

PLOS ONE